# Wireless Sensors for Brain Activity—A Survey

**Mahyar TajDini [1],\*, Volodymyr Sokolov [1] , Ievgeniia Kuzminykh [2,3] , Stavros Shiaeles [4]
and Bogdan Ghita [5]**

[1]   Department of Information and Cyber Security, Borys Grinchenko Kyiv University, 04212 Kyiv, Ukraine;
      v.sokolov@kubg.edu.ua
[2]   Department of Informatics, King's College London, London WC2R 2ND, UK;
      ievgeniia.kuzminykh@kcl.ac.uk
[3]   Department of Infocommunication Engineering, Kharkiv National University of Radio Electronics,
      61166 Kharkiv, Ukraine
[4]   School of Computing, University of Portsmouth, Portsmouth PO1 2UP, UK; stavros.shiaeles@port.ac.uk
[5]   School of Engineering, Computing and Mathematics, University of Plymouth, Plymouth PL4 8AA, UK;
      bogdan.ghita@plymouth.ac.uk
**\***   Correspondence: m.tadzhdini@kubg.edu.ua

**Abstract:** Over the last decade, the area of electroencephalography (EEG) witnessed a progressive move from high-end large measurement devices, relying on accurate construction and providing high sensitivity, to miniature hardware, more specifically wireless wearable EEG devices. While accurate, traditional EEG systems need a complex structure and long periods of application time, unwittingly causing discomfort and distress on the users. Given their size and price, aside from their lower sensitivity and narrower spectrum band(s), wearable EEG devices may be used regularly by individuals for continuous collection of user data from non-medical environments. This allows their usage for diverse, nontraditional, non-medical applications, including cognition, BCI, education, and gaming. Given the reduced need for standardization or accuracy, the area remains a rather incipient one, mostly driven by the emergence of new devices that represent the critical link of the innovation chain. In this context, the aim of this study is to provide a holistic assessment of the consumer-grade EEG devices for cognition, BCI, education, and gaming, based on the existing products, the success of their underlying technologies, as benchmarked by the undertaken studies, and their integration with current applications across the four areas. Beyond establishing a reference point, this review also provides the critical and necessary systematic guidance for non-medical EEG research and development efforts at the start of their investigation.

**Keywords:** brain wave; EEG signals; cognition study; brain-controlled games; NeuroSky; OpenBCI

## 1. Introduction

Recent innovative technology brought in an increasing proportion of hardware to the growing market for buyers of wireless, wearable single-channel, and multichannel electroencephalography (EEG) devices; given their miniature size and reduced price, such devices may be used regularly by individuals [1]. The traditional high-quality multichannel EEG devices are commonly used in medical or research fields in hospitals and laboratories, allowing clinicians and researchers to assess neural signatures in the patients and control different functions such as motor, sensory, and cognition [2]. From a data acquisition perspective, the excitation of neurons in the brain leads to a spontaneous electrical response, which can be recorded by electrodes from an EEG device [2–5].

However, traditional EEG systems need a complex structure and long period of application time, unwittingly causing discomfort and distress on the users [6]. While they may indeed lead

to noisier outputs and from narrower spectrum band(s), they also allow the continuous collection of data from regular environments, such as home or office. This allows their usage for diverse, nontraditional and non-medical applications, including cognition, brain–computer interface (BCI), and, further afield, education, gaming [7] assistance [8] and security [9]. There has been indeed a surge of interest in embedding brainwave data as an input into such applications, reflected in all segments of innovation [10,11]. A number of authors recently focused on determining how non-medical EEG can be used to enhance user interaction; this was naturally followed by manufacturers designing and building such devices, which were then integrated by developers within their applications. Given that consumer-grade EEG devices do not require standardization, unlike medical devices, as defined in EU directive 93/42/EEC (https://ec.europa.eu/growth/single-market/european-standards/harmonized-standards/medical-devices_en), and are not designed or intended to be used for diagnosis or treatment of disease, this area of research remains a rather incipient one, mostly driven by the emergence of new devices which represent the critical link of the innovation chain. In this context, the aim of this study is to provide a holistic assessment of the consumer-grade EEG devices, the success of their underlying technologies, as benchmarked by the undertaken research studies, and their integration with current application domains. Beyond establishing a reference point, this review also provides the critical and necessary systematic guidance for non-medical EEG research and development efforts at the start of their investigation.

The paper is structured into six sections. Following the introduction in Section 1, Section 2 provides an overview of the research methodology. Then Section 3 outlines a list of consumer-grade EEG products that have been used in past research studies. Section 4 critically reviews the research studies that investigated the efficiency, accuracy, and benefits of consumer-grade EEG devices. Section 5 discusses the impact of these devices on both traditional applications, i.e., cognition and brain–computer interfaces (BCI), and emerging applications, particularly educational research and game development. Section 6 concludes the paper with a set of achievements and findings.

## 2. Research Methodology

The paper aims to critically review the market of EEG consumer-level products and their suitability and prior usage within studies relating to brain activity. The discussion will focus in particular on the nature of the EEG signals recorded, the type of research that has been conducted and, where available, the performance of the consumer-grade device versus the medical-grade devices. In the context of virtual and augmented reality, assisted living, motor and neural-supportive devices, and IoT development, the requirements for EEG products are constantly evolving, both to make full use of the technological progress and the increasing customer demands. From an effectiveness and privacy perspective, these technologies are becoming more secure, accurate, have a quicker response time, and collect information without violating the legislation on the protection of personal data. This growing interest in the brain–activity technologies listed above has been matched by a similar increase in research activity in terms of both depth and breadth. A defining criterion of this study is to set aside the traditional research approaches relating to cognitive performance and BCI and focus on the applied research stemming from brain activity, such as education, assisting technologies, or gaming, with further analysis of the various research studies undertaken within each category. As part of the applied research, the analysis will particularly target consumer-grade devices and observe their performance versus medical EEG products.

Based on the tasks set, this study tackles the following research questions:

*RQ1*: What are the available EEG products that could be used for research, and what technical characteristics they have?
*RQ2*: What are the main application domains within the brain activity field?
*RQ3*: What type of research areas can be applied to each of the identified application domains?
*RQ4*: Which products were preferred for a certain research topic?

To answer these questions, we undertook a comprehensive review of the state of the art in the area of applied research for brain activity, focusing on studies using consumer-grade devices for data collection. We follow the review with a summary of the effectiveness of the identified devices and, where available, a comparative analysis with medical-grade devices.

## 3. Available Products

Over the last decade, noninvasive functional neuroimaging techniques, EEG in particular, have been introduced by research and embedded into solutions across a wide spectrum from academic research to commercial deployment. Brain signals recorded by EEG are interpreted and matched against the cognitive reaction and actions of subjects with the purpose of gaining a deeper insight into the functionality of the human brain. The neuroimaging devices, which are mostly adopted in medical-grade, require high accuracy and advanced functionality. In contrast, consumer-grade devices are rated high for their low cost, ease of setup, and for large-scale performance, which maybe makes them more appropriate for developing as a commercial product.

Currently available consumer-grade EEG systems have been reviewed by medical and clinical research in order to evaluate their accuracy, range, and data quality. An overarching conclusion is that they are not comparable with their medical-grade equivalents in terms of range, accuracy, or reliability, but they are also one or two orders of magnitude.

The analysis of the EEG consumer-grade devices identified several market leaders, namely NeuroSky, Emotiv and OpenBCI, that will be discussed in the following section. While they may vary in terms of specific characteristics and applications, all products have in common the objective to track brain waves and operate in the <100 Hz spectrum using various sampling rates. In addition, all products have applications for monitoring performance and are typically compatible with iOS and Android devices. The summative table of discovered products is presented in Table 1.

**Table 1.** Summary table of consumer-grade electroencephalography (EEG) products and built-in functionality.

| Product | Sensor | Channel | Sampling Rate [Hz] | Wireless Connection | Raw data Access | Operating Time (Up to, Hours) | Price [USD] | Released |
|---|---|---|---|---|---|---|---|---|
| NeuroSky MindSet *,1 | Dry | 1 | 512 | Yes | Yes | - | - | 2007 |
| Neural Impulse Actuator * | Dry | - | - | Yes | No | - | - | 2008 |
| Mindflex **,2 | Dry | 1 | 512 | No | No | - | 99 | 2009 |
| Emotiv EPOC *,3 | Wet | 14 | 128 | Yes | Yes | - | - | 2009 |
| MindWave *,2 | Dry | 1 | 512 | Yes | Yes | - | - | 2011 |
| XWave headset [12] | Dry | 1 | 512 | Yes | No | - | - | 2011 |
| Necomimi **,4 | Dry | 1 | 512 | No | No | - *** | 69 | 2012 |
| Emotiv EPOC+ 5 | Wet | 14 | 128/256 | Yes | Yes | 6 | 799 | 2013 |
| Melon HeadBand * [13] | Dry | 3 | - | Yes | - | - | - | 2014 |
| MyndPlay Myndband 4 | Dry | 1 | 512 | Yes | Yes | 10 | 219 | 2014 |
| Muse 6 | Dry | 4 | 220/500 | Yes | Yes | 5 | 199 | 2014 |
| OpenBCI 7 | Dry/wet | 8/16 | 250 | Yes | Yes | - *** | 199/950 | 2014 |
| Aurora Dreamband [14] | Dry | 1 | - | Yes | Yes | - | 299 | 2015 |
| Emotiv INSIGHT 6 | Semi-dry | 5 | 128 | Yes | Yes | 4 | 299 | 2015 |
| Muse 2 7 | Dry | 4 | 256 | Yes | Yes | 5 | 239 | 2016 |
| FocusBand [15] | Dry | 2 | 128 | Yes | No | 12 | 600 | 2016 |
| SenzeBand 8 | Dry | 4 | 250 | Yes | Yes | 4 | 299 | 2016 |
| MindWave Mobile 2 | Dry | 1 | 512 | Yes | Yes | 8 | 199 | 2018 |

* Not available, ** For entertainment, *** battery powered. 1 http://neurosky.com/, 2 https://store.neurosky.com/products/mindflex, 3 https://store.myndplay.com/products.php?prod=44, 4 https://www.necomimi.com/, 5 https://www.emotiv.com/, 6 https://choosemuse.com/, 7 https://openbci.com/, 8 https://www.neeuro.com/senzeband/.

### 3.1. NeuroSky

NeuroSky was one of the early implementors that designed, produced, and distributed consumer-grade EEG devices on the market, starting in 2007 with a NeuroSky MindSet. The hardware was hardly practical or portable, given it was a headset with two electrodes, one on the ear of the subject as a reference frame and the other one on the forehead for signal reading. The successful

release of MindSet led to the second generation of gadgets along with MindWave, MindWave Mobile, MindWave Mobile plus, and MindWave Mobile 2 (the only one still on the market). MindWave Mobile 2 appeared as an educational device and gaming application, and the key function was to monitor attention and meditation.

The devices from NeuroSky have been tested in different validation studies (Table 2). In 2016, Maskeliunas et al. [1] validated the main algorithms for monitoring attention and meditation and tested blinking recognition, concluding that MindWave can achieve only 22% overall recognition accuracy. This result is very low and shows its inability to discriminate between relaxation and focusing states. The blinking recognition algorithm provided by NeuroSky had 49.6% accuracy in the tests, which is why a gaze tracker (such as Eyetribe) would be preferable. The conclusion matches two previous studies: in [16], authors observed that MindWave had a high error rate of 43.52% when classifying whether eyes are opened or closed, while [17] identified the accuracy of blinking detection is only 40%. Despite the fact that blinking detection algorithm does not have a practical usage, the hardware by itself is effective enough, as established by Abo-Zahhad et al. by their study where was used raw eye blinking waveform MindWave created to identify person among 25 other people and achieved a 97.3% accuracy [18].

**Table 2.** Blinking recognition accuracy for NeuroSky device.

| Study | [1] | [16] | [17] | [18] |
|---|---|---|---|---|
| Accuracy | 49.6% | 43.52% | 40% | 97.3% |

### 3.2. Emotiv

Emotiv is another manufacturer of consumer-grade EEG devices, focusing on multichannel measurements with wet electrodes. Given their capabilities, Emotiv devices allow more complex tasks, such as creating and executing mental commands [19], or facial expression detection [20], 3D functional mind maps for data visualization [21]. Three Emotive systems reached the mainstream market: EPOC+, INSIGHT, and EPOC FLEX [22]. EPOC+ is a device focused on academic research, contains electrodes that are placed around the scalp and several built-in motion gyro, accelerometer, and magnetometer sensors. The company also released a cheaper alternative, called INSIGHT, with a lower number of channels, semidry electrodes, and was designed for everyday usage rather than for academic research. Finally, the lineup is completed by EPOC FLEX, a medical-grade product build with multiple sensors and other features, including a control box for wireless customization [22].

Consumer-grade products from Emotiv were used in different studies. For example, in 2013, Emotiv EPOC was used by Badcock et al. [23] to make research auditory event-related potentials (ERP) in adults and children [24]. The two studies were later followed by further ERP investigation using consumer [25] and medical [26] data acquisition procedures and equipment.

The EPOC system was repeatedly compared with medical-grade systems; Debener et al. represented some modifications to the Emotiv system and combined the hardware with a medical-grade EEG electrode cap [27] in 2012, then, in 2017, Barham et al. did the same modifications into Emotiv EPOC to evaluate it against a medical device [25]. Both ERP and MMN waveforms obtained from the Emotiv system shown similar results to the medical-grade device and to the results reaching in the Badcock et al. study [23]. The total data obtained from the modified gadget was higher than the one from the original report made by Badcock et al., mostly due to more accurate electrode placement and higher electrode quality.

Regardless of positive indications, there have also been reports about low-quality EEG data and poor performance regarding Emotiv EPOC. Duvinage studied the performance of Emotiv EPOC for P300-based applications [28,29] and against advanced neurotechnology (ANT), a medical-grade EEG system [30]. The overall performance and SNR of Emotiv EPOC were poorer than the ones produced by ANT. Similar to MindWave, the Emotiv EPOC system was used to validate attention and meditation results along with the blinking recognition [1]; Emotiv led to 75.6% accuracy for blinking recognition

and 60.5% accuracy for classification of focusing or relaxing. During the tests, only a small minority of subjects have been remarked a discomfort after a long period of gadget usage, but this was mentioned before in another research [31].

To summarize, when compared to medical-grade devices in scientific research data collection, the Emotiv EPOC system faces considerable challenges in data quality, sensor location, and electrode number. While not specifically highlighted by the prior studies, Emotiv EPOC does not need a flexible EEG cap. Therefore it does not require custom fitting; also, due to its form factor, it would allow the longer subject screening. The modified Emotiv EPOC [25,27], which was previously mentioned, has been proven as useful and also improved regarding competing against its competitors.

### 3.3. InteraXon

Muse, manufactured by InteraXon, is a headband with inbuilt brainwave sensors designed to facilitate meditation. In contrast to the products from NeuroSky and Emotiv, the fitting of electrodes from Muse can be adjusted to the head of the user. The most sensitive areas of the head are located under the Muse's electrodes, which help to obtain the emotion-specific signals [32]. In addition to measuring and monitoring the meditative state of the user, Muse also has a mobile application for meditation training (https://choosemuse.com/muse-app/). Following the launch of Muse, InteraXon expanded their product range with Muse 2, which offers more features (body movement, heart rate, and respiratory rate) and a software development kit (SDK) (https://github.com/NeuroJS/angular-muse), as well as a strong open-source base in GitHub (https://github.com/kowalej/BlueMuse).

In a study aimed at evaluating their accuracy in measuring ERPs, the Muse products [33] were used to obtain N200 and P300 ERP components in the oddball visual task and stimulate positivity during a reward-learning task. The results were compared to the signals resulted from Brain Vision actiCHamp System, which is considered to be the standard medical system. The study concluded that, while the performance of the two systems was relatively similar, the actiCHamp does deliver more accurate results. In the wider domain, there is a concern about data quality that Muse's configuration may not meet the criteria dictated by the guidelines for ERP's to study cognition (which includes specific requirements for formulation of the study, subjects profile, stimuli and responses, A/D signal conversion, signal analysis, non-cerebral artifact, presentation of data, and so on) [34], requiring one nonpolarized Ag/AgCl electrode for measuring the slow changes in potential and multiple electrodes which should note overlapping ERP components, as well as an adequate rejection ratio of the amplifier. The Muse headband is fitted with conductive silicon-rubber (dry) electrodes, which should be placed in four different locations. This access, focusing on the frontal part of the head, though not comparable with other consumer-grade EEG systems, allows data collection and validation, as presented by [33] and, with some adjustments, can produce higher quality signals.

In contrast to the medical-grade devices, the method to accurately label the time of experimental stimulations has been a major concern for researchers, as event markers are often needed to be created separately. In comparison to wired connections, Bluetooth is rather unstable and can introduce random delay variations [35,36].

The EEG data acquisition by Muse was compared to few medical-grade gadgets [6]. The data obtained from Muse shows an increase in broadband power and more signals, performance, and electrode placements, suggesting lower reliability. In addition, the data displayed by Muse has poor consistency between multiple measurements for the same subjects in comparison to other products (B-Alert, Enobio, and MindWave). The reason behind the inconsistency could be due to the mobile electrodes of the headband, which can be misplaced, an issue also confirmed by the conclusions of [33].

### 3.4. OpenBCI

OpenBCI was introduced to the market in 2013 as open-source hardware and firmware research alternative. Unlike consumer-grade EEG products, OpenBCI hardware mostly serves as an amplifier, where the activities detected in the brain (EEG), muscles (EMG), and heart (EKG) can be measured

by using the sensors equipped on the schematic OpenBCI printed circuit boards and interpreted using software named Processing. Due to its focus, OpenBCI has a limited market share and is more appropriate for users who have a background in engineering. A number of studies validated OpenBCI against traditional EEG systems. In 2016, Frey represented EEG signals obtained from OpenBCI's board versus a medical-grade EEG amplifier (g.tec g.USBamp) [37]. According to the obtained results, OpenBCI performance (measured using a combination of subjective, questionary-based, and objective, physiology-based indicators) was identical to the traditional EEG amplifier from g.tec for both P300 speller and EEG-based workload tracking by using the n-back task [38]. In a similar study [39], OpenBCI, equipped with Texas Instruments ADS1299, was compared to Compumedics Neuroscan NuAmps, a medical-grade EEG system original; the comparative analysis confirmed that the two systems produced similar results in terms of brain activity and frequency analysis. Regardless of these similarities, MRCPs are not normally benchmarked against the EEG applications, as they are designed entirely to help people with muscular disability to improve their quality of life and require movement, which can also cause incidents of cable swaying and undesirable head movements.

### 3.5. Other Products

Due to growing demand and rapid technology development, a wide range of non-research products were developed and released on the market. A BCI-based game controlling tool—Neural Impulse Actuator—was launched to replace the usage of a keyboard and mouse. Two other developers, X wave [12] and Melon [13], proposed products aiming to monitor user attention. More recently, a number of products appeared in the wellness industry by integrating the EEG data collection platforms: MyndPlay used NeuroSky TGAM technology to develop their headband; mostly, the company sells its apps. This application is usually used for entertainment purposes; however, it additionally touches brain training and educational areas. Aurora Dreamband [14], which is designed as a headband with built-in sensors, has the ability to detect the user's motion and track their sleep pattern. FocusBand [15] development was initiated in 2009 and finally was launched in 2016. In the beginning, the headset and app were manufactured with the goal to assist golfers to monitor and note their mental state and physical performance from the mentioned neurofeedback. To allow users to observe their performance in the working environment and sleeping condition, FocusBand expanded its targeted market of wearable headsets releasing NeuroSelf Care Business and NeuroSleep. The SenzeBand device is an EEG headband which has four channels and two reference sensors that can track the attention, focus, mental workload, and relaxation levels. Additionally, the company provides apps/games for mental training, managing stress, and keeping track of brain activities. The manufacturer of SenzeBand—Chief Scientist of Neuro—is an IEEE fellow, who as a professor at Nanyang Technological University, has been publishing numerous papers on neurofeedback/brain training game [7,40,41].

These low-cost devices can be counted as a tangible outcome of the advancement in medical technology and BCI, which has made a great leap into the consumer market. The lower price means increasing accessibility and realizability, which simplifies for researchers to study attention and subjects' mental state without going deeply into electronics and engineering questions. These devices may unlock several functions for public usage, be extended into education and gaming industries, but their functionality is limited in comparison to the medical-grade counterparts. Some grave disadvantages of these products from NeuroSky include low sensitivity because of the presence of a single channel in addition to inflexible location and electrode type. The EEG sensors of these gadgets responsible for reading attention and meditation conditions, due to their portability aspects, are likely to yield less accurate readings, but they can easily be adapted to evaluate student attention during a classroom or to trigger some events in the application. While NeuroSky was used in a small number of studies, its products have been used in many high-ranking publications, especially as part of educational studies. In contrast, Emotiv, with its 14 channels, brings better usage of signals while retaining its portability and was particularly preferred for validation studies. Muse is a rather new device, primarily aimed at meditation, but already being used in a wide variety of studies, especially if requiring a

larger number of channels. At the opposite end of the consumer-grade product, OpenBCI has high adaptability in a circuit board, requires prior understanding of electronics, bio-signal processing, and brain anatomy/physiology, but its full functionality and open source materials permit the users to develop and expand their applications and usage more freely and flexibly. Since it can be used with standard Ag/AgCl electrodes and accommodate up to 16 channels, the highest quality signals among the consumer-grade gadgets can technically be acquired using OpenBCI.

## 4. Analysis

In this section, we explore and summarize the research areas that have embedded EEG-based approaches in recent years. In the past decades, an investigation of the functionality of the human brain has considerably diversified from the health domain and physiological procedures to engineering practices.

### 4.1. Study of Cognition

Basic research in the field of cognitive psychology persists, considering the existing gaps in knowledge, but the field continues to work in the direction of unlocking the functional mechanisms of the brain. The concept of electric interaction in the brain attracted the attention of both neurologists and cognitive psychologists, who wanted to comparatively decode normal brain functionality versus neuropathological conditions [42,43]. Data on wave characteristics and their effectiveness to discriminate between regional or behavioral features, for instance, gamma waves (30–70 Hz) and the involvement in conscious perception, have gradually accumulated over the years [2]. Furthermore, brain–machine interface is on the leap, placing out to create a system capable of imitating human as closely as possible. If so, applied psychology for translational use has taken an investigative niche. Within the context of this review, many research groups have used the consumer-grade EEG gadgets in their studies, and this section of our work seeks to highlight the cutting-edge psychology-related research on emotion and cognitive functions that utilized consumer-grade EEG products, supplied by Emotiv, NeuroSky, InteraXon, and OpenBCI.

The study of cognition includes the following research areas:

- Emotions as emotion recognition and emotion classification;
- Attention and mediation;
- Mental workload as fatigue and stress;
- Memory capability.

The remainder of this section provides a broad summary of consumer-grade EEG studies on cognitive functions, together with sustained attention, working memory, and decision making. Table 3 summarizes the prior studies reviewed in this subsection and related to the study of cognition.

Emotional processing is a popular topic for scientists, from describing how emotion is recognized, stimulated in individuals to attempting to construct emotion recognition software. A typical example is provided by [44], which investigates the ability of the OpenBCI product in emotion recognition. This study was one of the first ones to represent evidence for the comparable performance of OpenBCI to high-end EEG derived data for emotion recognition algorithms. The data collected and analyzed with Open BCI and Empatica4 (E4) wristband led to the best prediction accuracy of 70% when using the K-means method, which indicates that the OpenBCI application could be extended to future research.

Other researchers described the usage of classification algorithms for affective tagging of audio-visual influence [45,46], albeit drawing EEG data using the Emotiv EPOC system. Katsigiannis and Ramzan in [47] used the support vector machine (SVM) method for the classification of devices based on EEG features, ECG features, or a combination of both. Liu et al. in [46] presented a real-time EEG-based emotion detection system that works on information transferred live-fed from the Emotiv EPOC headset using a 3-tier SVM-variant classification scheme. The stream through wire attaches to an EGGLAB toolbox in the Matlab for quick storage, which simplifies the signal processing, feature

extraction, and machine learning implementation. The reported accuracy varies because of different tiers, ranging from approximately 65% to 92%.

**Table 3.** Summary table of psychology-related research.

| Reference | Product | Method | Accuracy (%) | Subject | Study | Year |
|---|---|---|---|---|---|---|
| [48] | Emotiv EPOC | Independent component analysis | n/a | 27 | Emotion regulation | 2015 |
| [45] | NeuroSky MindWave Mobile | kNN+SVM | 63 | 20 | Emotion classification | 2016 |
| [47] | Emotiv EPOC | SVM | | 23 | Emotion classification | 2018 |
| [46] | Emotiv EPOC | SVM | 65–92 | 30 | Emotion classification | 2018 |
| [49] | NeuroSky MindWave Mobile | SVM | 86 | 15 | Emotion classification | 2018 |
| [44] | OpenBCI | K-means | 70 | 43 | Emotion classification | 2019 |
| [50] | Muse | Statistical analysis | n/a | 6029 | Attention and meditation | 2016 |
| [51] | Emotiv EPOC | Statistical analysis | n/a | 9 | Working memory | 2016 |
| [52] | Emotiv * | Statistical analysis | n/a | 18 | Working memory | 2016 |
| [53] | Emotiv EPOC | SVM | 65–67 | 19 | Subconscious face recognition | 2016 |
| [54] | Emotiv EPOC | Perceived relevance | - | 24 | Relevance judgement | 2017 |
| [55] | NeuroSky MindWave Mobile | SVM | 65–75 | 20 | Mental workload | 2017 |
| [56] | NeuroSky's chip (TGAM) | Statistical analysis | n/a | 15 | Mental fatigue | 2017 |
| [57] | Emotiv EPOC | CNN | 92 | 22 | Mental fatigue | 2017 |
| [58] | Muse | SVM | 65 | 209 | Esthetic preference | 2017 |
| [59] | Emotiv EPOC+ | Statistical analysis | n/a | 10 | Attention and vigilance | 2017 |
| [60] | Emotiv EPOC | RF, SVC, KNN, GP and 6 more | up to 80 | 86 | Attention and working memory | 2018 |
| [61] | Emotiv EPOC | Statistical analysis | n/a | 16 | Decision making | 2018 |
| [62] | Emotiv EPOC | SVM | 77 | 6 | Attention | 2019 |
| [63] | Muse | SVM | 92 | 50 | Drowsiness | 2019 |
| [64] | Muse | Binary classification | 64 | 28 | Mental stress | 2019 |

* The product name is not provided.

As well as emotion classification, a bit different line of research explores anxiety and stress quantification [45,49]. One more time, both lines use EEG data and machine learning techniques advantages. Zheng et al. [45] measured EEG and photoplethysmogram (PPG) responses in 20 volunteers using the MindWave Mobile headset and PPG-fitted glasses, respectively. To determine if EEG and PPG functions offered useful indicators for anxiety evaluation, the researchers used algorithms based on the principles of k-nearest neighbors (kNN) and SVM with radial basis function. On average, the classification accuracy was only 63%, clearly leaving some steps for improvement. Betti et al. [49] tested an SVM algorithm on 15 separated features from physiological measurements in the Maastrict acute stress test, such as the heart rate, electrodermal, and EEG signals (MindWave Mobile), and shown an 86% accuracy in releasing stress from a relaxed state.

One of the latest studies used the system for more traditional research into regional brain activation [48]. The participants of the experiment were involved in virtual reality (VR) simulation designed to set off sadness (e.g., poignant music, upsetting statements, or images), during which the concomitant brain activities were picked up with the Emotiv EPOC gear. The results showed that the Emotiv EPOC headset could be used as an investigative tool able to yield equivalently pertinent data to those which were received from complex neuroimaging systems.

Another field of cognition study is cognitive functions that include mental processes, e.g., attention, memory, and perception. One of such studies was using the Emotiv system and employed a combination of visual attention task and EEG brainwaves to decode attentional state to faces and scenes [62]. After the data from the participants was collected, a linear SVM classification was addressed on the recorded signals and was demonstrated that, on average, the prediction accuracy of behavioral performance was 77%, which seems on par with the previous fMRI study. Furthermore,

attention was applied in combination with mental exercise by Hashemi and colleagues from InteraXon in a large-scale investigation that compounded EEG recordings from 6029 subjects of all ages using the Muse headband [50]. Significantly, during the investigation were noted, discernible age-related changes in brainwave characteristics, which depended on sex and frequency band. Another study, conducted in 2017, called to monitor the workers' attention and vigilance with a wireless Emotiv product while they have been making an on-site object relocation [59].

Besides attention, working memory is another vital element of cognitive functions. After employing the Emotiv EPOC wireless headset, a group of researchers observed focused attention and working memory of the participants while they had been completing a battery of cognitive assessment tests related to 23 fundamental cognitive skills and 5 compound skills [60]. The extracted features from the collected EEG signals were fed to classifiers and reported that the constructed models were able to distinguish three levels (i.e., low, intermediate, and high) of focused attention and working memory with 84% and 81% accuracy, respectively. In a similar way, some other research group used a well-known dual N-back task, which was designed to tap into a person's working memory capacity [52]. The results showed that different memory workload levels corresponded to the reaction time of the participants and accuracy and manifested as disparate EEG patterns.

Buszard et al. in [51] used Pearson's automated working memory method to investigate the connection between working memory capacity, EEG coherence, and performance under pressure. The results of the experiment confirmed the correlation between verbal-analytical process, brain activity and motor performance. In total, individuals with larger verbal working memory ability exhibited greater verbal involvement (explicit motor learning) while motor performance, whereas larger visuospatial capacity, was linked to reduced verbal involvement (implicit motor learning).

Other studies of cognitive functions include research of relevance judgement (Emotiv) [54], decision making (Emotiv) [61], mental workload (NeuroSky) [55], mental fatigue and stress (NeuroSky, Muse, and Emotiv) [56,57,63], subconscious face recognition (Emotiv) [53], and esthetic preference (Muse) [58].

Ramsy et al., in [61] investigated a person's willingness to pay and brain activity. Recorded by Emotiv EEG signals showed that gamma and beta band asymmetry in the prefrontal areas of the brain corresponded to the willingness to pay. Frontal EEG was additionally put to harnessed for mental workload evaluation during four cognitive and motor tasks, which were calculation, finger tapping, mental rotation, and lexical decision [55]. A concomitant increase displayed theta activity while the task was becoming more difficult, and an SVM-based model was 65% to 75% during mental workload associated with distinctive activities.

Morales et al. investigated in [56] brain activity metrics as EEG power spectral density and saccadic peak velocity during fatigue after riding. The records were done using a ThinkGear ASIC module (TGAM) headset (NeuroSky) together with saccadic velocity sampled with infrared oculography (JAZZ-novo). To score fatigue Stanford sleepiness scale and an adapted version of the Borg rating of perceived exertion was used. The results showed that the TGAM headset could be used as a detection system for mental state modifications that occur during day-to-day task execution, for example, driving. Another recent research [63] used Muse for detecting and predicting drowsiness by data from multiple sensors, i.e., accelerometer and gyroscope. Using a combination of the EEG spectral analysis, blinking detection from EOG, head movement from gyroscope and accelerometer, and SVM as a classifier, the researchers carried out 92% accuracy for predicting whether the participants were alert or drowsy.

The effectiveness of stress detection using EEG data was investigated in [57,64]. Muhlbacher-Karrer et al. in [57] used Emotiv product to collect metrics related to a driver's state as electrodermal activity, electrocardiogram, and capacitive hand detection sensor. The data were used to extract an average of 25 features for a channel and fed to cellular neural network classifiers (CNN). The results showed that EEG data were informative in the detection of stress, which allowed us to achieve an accuracy of 92%. The authors in [64] used Muse to record EEG data from 28 participants in the condition of pre- and post-activity. The results showed that binary classification (stressed vs. non-stressed) with their novel

feature selection algorithm had a higher prediction accuracy as compared to the 3-class alternative (93% and 64%, respectively).

Further research in the field of cognition study investigated the possibility to investigate the subconscious mind via facial recognition [53] and to identify the individual's artistic preference [58]. The first work used the Emotiv EPOC headset for collecting signals during seeing of well-known people and SVM classifiers to analyze the data. Results showed an average accuracy of 65%. Second work used Muse headbands to measure EEG correlates of a group of volunteers while they had a tour of the art exhibition. The researchers found out that the piece of art that was favorite among participants caused brain activity out of the baseline condition compared to the reaction to other paintings.

In conclusion, the development of the usage psychology-related consumer-grade EEG app seems to suggest that in psychological research, using consumer-grade devices is similarly distributed between the investigation of emotion, attention, and mental processing. Additionally, in terms of the prevalent product brand, Emotiv is leading. In addition, the most interesting is that the biggest part of research mostly focused on the application of brainwave indicators as a proxy for interpretation by classification algorithms, more particularly supervised machine learning techniques.

### 4.2. Brain–Computer Interface

One of the major application fields for EEG equipment is the brain–computer interface (BCI). BCI has been undergoing consistent innovation and improvement since the 1970s [65,66]. This led to a number of products, such as the P300 speller, produced in 1998, which allowed people with motor disabilities to pick alphabets on a computer screen through visual perception and brain responses [67]. The P300 wave is an event-related potential (ERP) event that is usually triggered by visual, auditory, and tactile stimulation. While relatively effective, the P300 speller had a prohibitive price tag; hence it never reached the consumer market. In recent years, several P300-BCI related studies have been undertaken. Some of them chose the OpenBCI platform, particularly due to its flexibility [68,69], others preferred the applications of Emotiv [69,70], and in [71], the authors analyzed both systems. A common thread of the research was to provide more user-friendly practical applications. Table 4 below provides a summary of the studies.

The steady-state visual evoked potential (SSVEP) represents a type of visually evoked brain response, first integrated into a BCI system in 2000 [72]. The device aims to attribute and correlate a brain signal, as recorded by an EEG amplifier, to a specific visual stimulus frequency, such as a constant flickering stimulus on a screen. Later on, subsequent products focus on multiple frequencies and using canonical correlation analysis for discriminating between them [73].

**Table 4.** Summary of brain–computer interface (BCI)-related research studies that were reviewed.

| Reference | Product | Method | Accuracy (%) | Subject | Research Topic | Year |
|---|---|---|---|---|---|---|
| [68] | OpenBCI | Statistical analysis | 61–83 | 7 | ERP | 2015 |
| [74] | OpenBCI | Statistical analysis | 83–86 | 10 | ERP | 2016 |
| [69] | Emotiv EPOC+ | LDA, SWLDA, BLDA, SVM, NN | 75–92 | 14 | ERP | 2017 |
| [70] | Emotiv EPOC | Least squares, FLDA | 85 | 6 | ERP | 2017 |
| [75] | OpenBCI | Unknown | 66 | 4 | ERP | 2018 |
| [71] | Emotiv EPOC | SVM | 80 | 10 | ERP | 2018 |
| [76] | Emotiv EPOC | canonical correlation analysis | 78 | 10 | SSVEP | 2018 |
| [77] | Emotiv EPOC | MLP NN | 96–99 | 5 | SSVEP | 2018 |
| [78] | OpenBCI | Statistical analysis | 50–92 | 4 | SSVEP | 2018 |
| [79] | Emotiv EPOC | Statistical analysis | n/a | 30 | MI | 2017 |
| [80] | Emotiv EPOC | ANN | 75 | - | MI | 2018 |
| [81] | OpenBCI | Quadratic discrimination analysis | 83–90 | 1 | MI | 2018 |
| [82] | OpenBCI | MLP NN | 64–85 | 7 | MI | 2018 |
| [83] | OpenBCI | FLDA | 85–99 | 10 | Other | 2017 |
| [84] | Emotiv EPOC+ | RF | 88–96 | 60 | Other | 2018 |
| [85] | Emotiv EPOC+ | BLSTMLSTM | 94–98 | 60 | Other | 2018 |
| [86] | OpenBCI | Statistical analysis | n/a | 3 | Other | 2019 |
| [87] | OpenBCI | Statistical analysis | n/a | 20 | Other | 2020 |

A limited number of studies focused on the usage of consumer-grade EEG for monitoring SSVEP-BCI. Authors in [76] successfully linked visible feedback using a combination of Emotiv and a head-mounted device named HTC VIVE to familiarize disabled people with brain–machine interaction on a 3-D space. In two separate studies, Lamti et al. developed EEG and gaze data fusing framework for wheelchair navigation [77] and, in [88], Emotiv and a Tobii eye tracker (EyeX model) that were were used to monitor brain and eye activities.

Auditory steady-state responses (ASSR) are a class of steady-state EEG responses stimulated by using constant auditory stimulation (regular tone) [89]. While a participant listens to a speaker-generated constant sound, that individual's EEG can be varied at an equal tone or frequency (focusing on the temporal region in particular [90]). ASSR-BCI was integrated into OpenBCI-based platforms for a home automation system allowing the user to control the devices completely with the auditory signals [78]. Kaongoen et al. followed with a hybrid system, including ASSR and P300, also built on the OpenBCI platform [83]. The research demonstrated that using both stimuli of ASSR and P300 simultaneously could lead to an improvement inside the performance classification of a selective attention task and reach 99% accuracy.

Motor imagery (MI) was also used as part of EEG as an innovative BCI avenue [91]. The MI response was generated by activating the neural correlates of motor functions without actual motor execution. The most common tasks for MIBCI research are the imagery of the left/right upper and lower limbs in addition to their functional performance. Many previous studies used ultramodern experimental protocols with the datasets to conduct and analyze various imagery tasks (http://www.schalklab.org/research/bci2000) (http://bnci-horizon-2020.eu/).

The adaptation of the consumer-grade EEG device was mentioned in a study [80] that used a version trained by large MI-EEG datasets got from the medical-grade device as the operating model for unseen data recorded from Emotiv. The model got acceptable results as a wheelchair controller. This research brings into focus the benefits of large-scale public datasets run from the medical-grade device in enhancing the performance of those from the consumer-grade device. To demonstrate the benefits of consumer-grade (OpenBCI), the study in [81] collected a large amount of MI-EEG contained images right/left hand moving and resting tasks to investigate response accuracy; the study was further extended in [82] to compare OpenBCI with medical-grade, with a DNN processing.

A different approach was taken in [92,93], which focused on the quality of the controls and inputs of BCI, more specifically connectivity (such as portable/wearable, wireless, dry electrodes, electrode

montage, etc.), process (including online, single-channel, artifact removal especially eye blink, etc.) and environmental interaction (performed outdoors, the performance of daily life tasks, etc.).

The conclusion was that, apart from the actual strength of the BCI signals, the resulting performance is a combination of the inputs implementation, signal processing, and noise introduced by the surrounding context.

In addition to the above mentioned BCI-based studies, categorized as the research of ERP, SSVEP, and MI, a separate effort was directed towards the areas where consumer-grade EEG devices are used for both regular and medical usage. In 2018, [84] proposed the use of the relaxation state towards the prediction of age and gender while using EEG-BCI; the results can support various applications, which include biometric, healthcare, entertainment, and targeted advertisements. The study used the Emotiv EPOC+ and achieved 88% accuracy for age and 96% for gender classification by using the random forest algorithm. In a subsequent study [85], Kaushik et al. embedded a deep learning approach named BLSTMLSTM (a combination of bidirectional long short-term memory (BLSTM) and deep long short-term memory (LSTM)) on the same datasets and reached up to 94% and 98% for age and gender classification, respectively.

As indicated by some of the above-mentioned studies, BCI processing has recently embedded computational intelligence, deep learning (DL) to support the signal analysis and decision-making process. A new adaptation of the leading technologies can significantly reduce the cost and eliminate the mobility barrier for collecting large-scale EEG-based datasets. The cycle of data feeding into the DL algorithm can bring to the future development of applications to improve the quality of life.

### 4.3. Educational Research

Educational researchers prefer minimum weight, easy-to-use gadgets because of their low price, wear-resistance, single-channel, and dry electrodes. Such devices offer simpler solutions to monitor brain activity while the participants are busy with various learning tasks.

Three research areas are of substantial significance in the education-related EEG applications: attention and meditation, engagement time, and brain-to-brain synchrony.

As shown by the analysis of reviews presented in Table 5, most of the undertaken research focused on the study of attention, in spite of the fact that there are many cognitive aspects to be explored for improving the educational environment. To further support the users, Emotiv now validates the performance metrics such as stress/frustration, engagement, interest/valence, excitement, focus/attention, and relaxation/meditation (https://www.emotiv.com/knowledge-base/performance-metrics/). Most of the studies would also welcome a more statistically confident approach, as they tend to be based on small subject groups. Given the existing breadth of research and level of investigation, it can be concluded that educational research based on EEG is still in its early stage in terms of both technology and knowledge.

The study of attention is the most popular research area that could be seen from Table 5. The methods for attention recognition and evaluation are mostly developed to measure the effectiveness of various teaching techniques in educational research and varied due to the feature extractions or denoising algorithms such as the usage of spectral features, principal component analysis (PCA), and multi-wavelet transform. Current studies are leaning towards machine learning as a classifier with a mixture of SVM, Bayesian classifiers, Markov models, k-nearest neighbors, and neural networks. The accuracy of the categorization is usually more than 75%, depending on the tasks and experimental installation.

In [94], Liu et al. collected raw EEG signals from 24 people with a NeuroSky MindSet to evaluate the level of attentiveness. Totally, there were five features extracted, such as the energy value of every frequency, together with the ratio of alpha and beta activities and trained with SVM classifiers. As a result, the study reached up to 76.82% classification accuracy and showed that the alpha activity is associated with a relaxed state and the increasing of the beta activity means an increase in attention.

Since 2014, many educational studies are counting just on EEG devices as a validation tool. These studies have been using NeuroSky's algorithm to monitor the subjects' level of attention while they are doing the learning-related tasks. Attention level of students was investigated in [95] using genetic algorithm for feature selection together with SVM classifier. Mobile polling, as a means of interactive learning, was studied in [96]. The level of attention while using mobile polling may be lower compared to the traditional clicker, but it can increase during the activity. In an experiment with the affection of displayed text on the user's attention, researchers Che and Lin [97] have confirmed that distinct forms of text display, named static, dynamic, and mixed, do not show any tremendous results at the users' attention. In [98,99], the effect of different genres of books on individuals from distinctive age groups (elementary school pupils and adults) were explored. Results showed that stimulating audiobooks or e-books are recommended to be suitable for elementary school boys, while conventional books are better for grade elementary school girls.

Additionally, to measure the maintained attention level, Chen and Wu in [100] have exercised the emWave system to detect emotion, cognitive load, and learning performance in subjects. EmWave affords software to monitor heart rhythm and variability-based emotion recognition algorithm of heart rate. A separate direction of research, outlined in [95,99], used FaceReader, a facial expression detection software, to point the emotional states together with EEG-based attention level of the subjects. The results showed that the students, who obtained the rewards for the correct answers, paid more attention than the ones who received no rewards. Anyway, no connection has been determined between the positive stimuli and the learning performance.

**Table 5.** Summary of education-related research studies that were reviewed.

| Ref. | Product | Method | Accuracy (%) | Subject | Research Topic | Year |
|---|---|---|---|---|---|---|
| [94] | NeuroSky MindSet | SVM | 77 | 24 | Attention | 2013 |
| [96] | NeuroSky MindSet | Questionnaire | n/a | 32 | Attention and meditation | 2014 |
| [101] | NeuroSky MindSet | Questionnaire | n/a | 20 | Attention and meditation | 2014 |
| [102] | NeuroSky MindSet | Questionnaire | n/a | 126 | Attention and meditation | 2014 |
| [103] | NeuroSky MindSet | Statistical analysis | n/a | 5 | Attention and meditation | 2014 |
| [100] | NeuroSky MindSet | Statistical analysis | n/a | 10 | Attention and meditation | 2015 |
| [99] | NeuroSky MindSet | Questionnaire | n/a | 96 | Attention and meditation | 2016 |
| [97] | NeuroSky MindSet | Questionnaire | n/a | 20 | Attention and meditation | 2016 |
| [104] | NeuroSky * | Statistical analysis | n/a | 42 | Attention and meditation | 2016 |
| [105] | NeuroSky MindWave Mobile | Questionnaire, interviews | n/a | 30 | Attention and meditation | 2016 |
| [95] | NeuroSky MindWave | SVM | 75 | 10 | Attention and meditation | 2017 |
| [100] | NeuroSky MindBand ** | Statistical analysis | n/a | 78 | Attention and meditation | 2017 |
| [106] | NeuroSky MindWave | Post-test, interviews | n/a | 60 | Attention and meditation | 2017 |
| [107] | NeuroSky * | Questionnaire | n/a | 44 | Attention and meditation | 2017 |
| [108] | NeuroSky MindWave | Questionnaire | n/a | 60 | Attention and meditation | 2017 |
| [109] | NeuroSky MindSet | Questionnaire, interviews | n/a | 20 | Attention and meditation | 2017 |
| [110] | NeuroSky MindWave Mobile | Questionnaire, post-test | n/a | 80 | Attention and meditation | 2017 |
| [111] | NeuroSky * | Pre-test, posttest | n/a | 148 | Attention and meditation | 2018 |
| [112] | Emotiv EPOC | Questionnaire | n/a | 48 | Engagement time | 2014 |
| [113] | Emotiv EPOC | Statistical analysis | n/a | 50 | Engagement time | 2016 |
| [114] | Emotiv EPOC | Post-test | n/a | 12 | Brain-to-brain synchrony | 2017 |
| [115] | Emotiv EPOC | Questionnaire | n/a | 12 | Brain-to-brain synchrony | 2019 |
| [116] | Emotiv EPOC | Linear regression, kNN, SVM | 46–63 | 10 | Other | 2011 |
| [117] | Emotiv EPOC | Statistical analysis | n/a | 40 | Games in education | 2015 |
| [118] | OpenBCI | Statistical analysis | n/a | 22 | Other | 2018 |
| [119] | Muse | Statistical analysis | n/a | 26 | Other | 2019 |

\* The name of the product is not provided. ** NeuroSky's research and development only product.

Although the biggest part of the researchers is focusing just on attention point, some of them also include meditation into their studies as NeuroSky also supplies its pride meditation detector directly to the products [96,104,106,108]. However, meditation will not play the main role in comparison to attention during an active classroom and can only be included with other factors for performance evaluation.

Along with the usage of the consumer EEG products for evaluation of the teaching/learning methods, there are also researchers who made one more step further and developed different systems for maintaining the students' attention and improving their learning performance. These systems usually display a user's attention level and initiate a set of acoustic alarm when their attention drops beneath a certain threshold [109–111]. They also can create an interactive agent that has the capacity to give proper feedback based on the user's emotion and attention levels [105].

The task of measuring the level of engagement is more complex than evaluating attention, may depend on the individual's satisfaction, feeling of freshness or usability. This is one of the reasons why it is important for the learning process to have face-to-face classes or online lessons. In 2014, a new sensor-based method for analyzing the user's motivation was proposed in [112] by combining few metrics based on the time of interception applied by Affectiv Suite's EmoEngine (old name of an algorithm provided by Emotiv). In addition, it is real time and can provide a deeper perception of the change in the motivation level. Authors in [113] suggested a mechanism called time on task threshold computation (ToTCompute). The goal of this new mechanism was to help educational game creators monitoring the participant's engagement during their interactive lessons. The system used performance metrics collected from Emotive to automatically count engagement level, and in case of low engagement value to triggering special events to motivate students.

The brain-to-brain synchrony has been developed to study social dynamics, which include the interaction among students and teachers in the real classroom and show a relationship between the brainwaves of multiple individuals related to each other. The researchers in [114,115] collected EEG signals from the students and their teachers in the regular classroom and computed synchrony. There were monitored brain-to-brain synchrony of the student-to-group, student-to-student, and student-to-teacher interactions. The results in [114] showed that student-to-group synchrony was more related to teaching techniques (video or lecture), but between classmates, students showed the highest student-to-student synchrony. The results in [115] showed that synchrony of the group of students who were studying with videos was higher than another one with lectures. The student-to-teacher synchrony showed their closeness. The results confirmed how cognitive outcomes, such as the student's academic performance, can be predicted by social dynamics, like perceived closeness.

Other directions in investigating brain activities for education-related purposes are the classification of cognitive loads, predicting the customer frustration, the possibility estimation that the user will fail the test, detection of language comprehension and language transition in Duolingo application [120,121] detection of suspicious behavior during exam using EEG and eye-tracking.

The cognitive load during reading was analyzed in [119] using Muse and EyeTribe eye-tracker for data collection. The experiment showed that brain activity during the reading of more difficult contents and easier contents is different. It displayed the influence of the textual contents on the language processing related to the left temporal region. In addition, the study analyzed the eye movement and its pattern, and the results showed the acceleration while the easier contents were reviewed, affecting the brain activity of the left frontal lobe.

The level of frustration and excitement may have an enormous influence on the students learning ability. The researchers in [116], using Emotiv EPOC, tried to predict students' emotion after received feedback from the Intelligent Tutoring Systems, a computer system able to provide feedback for cognitive assistance. The collected data were analyzed with several machine-learning algorithms: linear regression, kNN, and SVM to predict the user's emotion level. The outcomes suggested that linear

regression plays with higher quality than the other algorithms with R2 of 0.462, 0.560, 0.627, and 0.484 for predicting frustration, excitement, changes in frustration, and changes in excitement, respectively.

Gamification in education became very popular and identified as marketable for researchers and developers. The authors in [118] analyzed and tried to predict the success or failure of the students in each of the tasks in the game using EEG signals from Emotiv EPOC (short/long-time excitement, mediation, frustration, and tedium), the diameter of the pupil from an eye tracker to evaluate the participants' workload, valence, and arousal, and facial expression from FaceReader software. The game is programmed to detect if the player is progressing with the game and offer various examples or hints for better understanding. The research successfully carried out 66% accuracy with logistic regression, a machine learning model.

*4.4. Gaming*

From the application perspective, consumer-grade EEG devices also draw the attention of researchers in the entertainment and media sphere, especially in gaming. A number of studies, outlined in this section, aimed to investigate the ability of EEG devices to capture the feelings [122] and the emotions [123] of the gamers during a game by reading the activity of the brain. The recording of the brain activity has also been used to classify the level of expertise of the players involved in the game [124] or to analyze the emotional process experienced by the patients with neurological and neurodevelopmental issues [125]. The above works can be divided into several groups that study the:

-   Effect on neurofunction (as emotions);
-   Brain-controlled games;
-   Neurological disorders and improvement.

There are many types of games, including mobile games, video games, and computer games; the games have been designed mostly for education and training, as well as serious games. It has been found that video games can relax players, bring relief from stress, improve skills, but also may be harmful and destructive to the gamers' health and wellbeing. The outcomes of gaming can be divided into two groups, named effects of emotions and experience. The study [122] investigated the effects of games on emotion using the Emotiv device together with saliva samples from the gamers to identify the level of stress created by the video games. Results showed that the fear and violent excitement games had increased a level of stress, and, therefore, both could have a destructive influence on the players' health. From the other side, there was no stress after playing the puzzle game, and it was little after the runner game.

Different methods have been introduced regarding the brain functions during the playing games with the use of consumer-grade EEG devices [122,126].

A method suggested by Mondjar in [127] provides the analysis of the recorded EEG signals, which showed correlation with specific mechanics of playing games, which affect the cognitive function of a player. The results showed that serious games could stimulate gamers to exercise the brain areas responsible for memory, attention, and concentration. It was determined that these cognitive activities were stimulated via five mechanics: accurate action, timely action, pattern learning, a logical puzzle, and mimic sequences.

A way to record emotions, analyze them, and determine the correlations to the signals was validated using the Emotiv EPOC headset [122]. Compared to the previous studies, during the gameplay, the recorded brainwaves showed changes in brain activity. The results showed that various scenes during the gameplay caused differing types and intensities of emotion in each player, for example, the excitement from correctly hitting the target and frustration from missing the goal [128]. For visualization of the brain activity have been used two colors: the area of the brain that performs high activity used red color, and the area with low activity used green color to present emotions during the game [126,129]. Furthermore, many neurogaming methods have funded the research of affective states on the task engagement and the 2-Dimensional valence/arousal plane [130]. These

studies aimed to help in the adaptation of various types of games for patients with neurological and neurodevelopmental disorders for clinical evaluation of emotion.

In several studies, the EEG products were used to categorize the expertise level of a player during the game based on brain activity [124,131,132]. The first two works analyzed collected data and tried to predict the level of performance of each player (as professional or amateur) using naïve Bayes and SVM Classifier. Stein in [132] launched research regarding the classification of players' competencies and dynamic difficulty adjustment (DDA) with the help of the Emotiv system. The EEG signals were analyzed with the goal of discovering the excitement level experienced by the players. DDA means adjusting the game by reducing difficulty for weaker players and increasing difficulty for stronger players. Even more, an approach that uses the EEG signals as a base for adaptive game-based learning has been launched [133]. The brain activity was continuously monitored, and the game mode was very fast adjusted when the decrease in the excitement level was detected in the position below a predefined threshold.

In comparison to the traditional game in which the players physically control the game avatar using a keyboard, mouse, joystick, foot paddle, brain-controlled games suggest an unconventional approach for gamers. Special headsets are capable of detecting the changes in brainwaves and give the gamers the possibility to concentrate only on the game. The idea of attention-based mind-controlled games is gaining the attention of not only game producers but also research companies [134–136]. In 2018, Queiroz et al. [136] launched a BCI-based game where the player has control via a wheeled robot through the Emotiv INSIGHT headset to reach a target. The headset was tuned up to software, permitting the conversion of the commands as hot or cold through the computer interface controlling the movements of the robot-like spinning around or changing direction. In the same way, Vasiljevic in [134] and [135] launched an attention-based BCI game named Mental War, represented as a tug-of-war game. The value of attention captured by the NeuroSky MindWave headset and read during some period of time have been calculated for both players. The higher attention value specifies the strength that use the avatar to pull the rope and win towards the opponent.

Neurological and neurodevelopmental disorders can seriously influence the patients' everyday life, in particular on mental stability. Patients who have these types of disorders facing all types of problems related to health, academic, social relation, and occupation during their lives. Neurofeedback technology is one of the behavioral, non-pharmacological remedies, which has started to become popular because of its promising effectiveness in disorder treatment and the improvement of the patient's health. Many studies [125,137–146] presented the concept of neurofeedback training and showed that it gave a treatment option for patients, a decreasing in the fundamental symptoms such as in the cases of cerebral palsy [147], stroke [148], and ADHD [149–152]. Generally, the treatment consists of the patient's request to wear a BCI headset and perform tasks specifies by a trainer. The headset records the brain signals and transforms them into visual signals for real-time feedback. For any successfully finished trial, for example, performing a suggested task correctly, there is a reward that is given for motivation to inspire the patient.

Often for treatment in conjunction with EEG headsets, the so-called "serious games" are used, a type of video games called in this way due to their special characteristics include such tasks beyond just leisure purpose, to detect any abnormal features and improve problems with attention [129,142]. Diverse mind-controlled games, "RehabNet" [148], "Harvest Challenge" [138], "Shooting" [139], "Magic Carpet" [143], "FOCUS" [142], "MindLight" [143,144], and "FarmerKeeper" [146], are instances of video games and serious games, which serve as neurofeedback training for patients that have cognitive issues, wherein the characters are controlled by signals changed in the brain and their patterns. All these games promote a similar goal, for example, to enhance the cognitive function of the patients with mental issues and serve as a kind of training therapy for the patients recovering from neuro-related disorders, such as ADHD, cerebral palsy, dementia, paralysis, stroke, etc. By showing the patients their brainwave patterns, it is possible to increase awareness of the specific changes in their physique that generally are not consciously controllable.

In this section, numerous factors of EEG applications have been discussed in the gaming domain: detection of emotion, gameplay media, and patient rehabilitation. and summarized in Table 6. For now, the growing necessity of BCI for various purposes in gaming has provoked huge development of the technology. However, for future studies, it is very important to evaluate the effectiveness of the gaming system with the purpose of amplifying the data processing and hardware development for the ultimate improvement in players' experience.

**Table 6.** Summary table of gaming-related research.

| Reference | Product | Method | Accuracy (%) | Subject | Study | Year |
|---|---|---|---|---|---|---|
| [130] | Emotiv EPOC | Statistical analysis | n/a | 30 | Effects on neurofunction | 2015 |
| [131] | Emotiv EPOC | Naive Bayes, SVM, MLP | 80–89 | 10 | Effects on neurofunction | 2016 |
| [153] | NeuroSky MindWave Mobile | Questionnaire | n/a | 20 | Effects on neurofunction | 2016 |
| [128] | Emotiv EPOC | Statistical analysis | n/a | 20 | Effects on neurofunction | 2016 |
| [126] | Emotiv EPOC | Statistical analysis | n/a | 12 | Effects on neurofunction | 2016 |
| [127] | Emotiv EPOC | Questionnaire | n/a | 10 | Effects on neurofunction | 2016 |
| [122] | Emotiv EPOC | Statistical analysis | n/a | 80 | Effects on neurofunction | 2018 |
| [123] | Emotiv EPOC | Statistical analysis | n/a | 15 | Effects on neurofunction | 2018 |
| [124] | Emotiv EPOC | Naive Bayes, SVM, MLP | 82–86 | 20 | Effects on neurofunction | 2018 |
| [132] | Emotiv EPOC | Questionnaire | n/a | 8 | Effects on neurofunction | 2018 |
| [133] | NeuroSky MindWave Mobile | Post-test | n/a | 8 | Effects on neurofunction | 2018 |
| [129] | Emotiv EPOC | Statistical analysis | n/a | 10 | Effects on neurofunction | 2019 |
| [135] | NeuroSky MindWave Mobile | Statistical analysis | n/a | 16 | Brain-controlled game | 2018 |
| [136] | Emotiv EPOC | Statistical analysis | n/a | 2 | Brain-controlled game | 2018 |
| [134] | NeuroSky MindWave Mobile | Statistical analysis, Interviews | n/a | 8 | Brain-controlled game | 2019 |
| [148] | Emotiv EPOC | Questionnaire | n/a | 3 | Neurological disorders | 2013 |
| [137] | Muse | Statistical analysis | n/a | 577 | Neurological disorders | 2015 |
| [125] | NeuroSky * | Questionnaire | n/a | 160 | Neurological disorders | 2015 |
| [138] | NeuroSky MindWave | Statistical analysis | n/a | 9 | Neurological disorders | 2016 |
| [139] | Emotiv EPOC | Statistical analysis | n/a | 5 | Neurological disorders | 2016 |
| [140] | Emotiv EPOC | Statistical analysis | n/a | 107 | Neurological disorders | 2017 |
| [141] | Emotiv EPOC | Statistical analysis | n/a | 9 | Neurological disorders | 2017 |
| [147] | Emotiv EPOC | Statistical analysis | n/a | 8 | Neurological disorders | 2017 |
| [142] | Emotiv EPOC | SVM | 98 | 5 | Neurological disorders | 2018 |
| [143] | NeuroSky * | Statistical analysis | n/a | 174 | Neurological disorders | 2018 |
| [144] | NeuroSky * | Statistical analysis | n/a | 43 | Neurological disorders | 2018 |

* The name of the product is not provided.

## 5. Discussion

From the analysis conducted in the previous sections, we summarize the following key findings with regards to the research questions posed in Section 2 concerning the review of studies in the area of applicability the consumer-grade EEG products.

*RQ1: What are the available EEG products that could be used for research, and what technical characteristics they have?* A comprehensive review was provided for the leading products, based on the existing, published research studies. Given the emphasis for these devices is placed on the application domains rather than obtaining accurate readings, information about their technical characteristics is somewhat limited, describing the measurement range rather than the sensitivity or error levels. Currently, products from the four main manufacturers, called, NeuroSky, Emotiv, InteraXon, and OpenBCI, of consumer-grade EEG devices are available for purchase. Among the four, NeuroSky products appear the most successful in seizing the market due to lower prices, simplicity, and ease of use, including a single-channel EEG sensor. Unlike NeuroSky, Emotiv devices focused on the cognition and

gaming fields due to two main reasons—data collection across 14 channels and being more sensitive than NeuroSky, providing it with the ability to simultaneously examine more brain regions and signals. An interesting alternative to EEG devices has been the one provided by Muse. Primarily designed for meditation, recent Muse devices have an accelerometer, heart rate, and respiratory rate sensors built-in, all of which made the product more attractive and allowed consumers to monitor more physiological factors. Comparing to the alternative ready-to-wear devices, OpenBCI offers a less user-friendly product, which is essentially a circuit board. Its open-end design (open-source hardware, firmware, and GUI) made it very appealing for engineering-related research studies in the BCI field. The focus for OpenBCI has been indeed the acquisition and processing of signals rather than its usability. The full list of the products with their characteristics is presented in Table 1.

*RQ2: What are the main application domains within the brain activity field?* A wide variety of research that relied on EEG-based devices to investigate human brain activity was identified, ranging from health and wellbeing-support implementations to enhancing processes relating to engineering. The summarized studies were clustered around four major areas: (1) study of cognition, (2) brain–computer interface-related, (3) education-related, and (4) gaming-related studies.

*RQ3: What type of research areas can be applied to each of the identified application domains?* Each one of the four application domains identified by RQ2 has specific research topics, summarized in Tables 3–6. The summary shows the model of the EEG product that was used in research, what research method was used to obtain results, the number of subjects that participated in the experiment, the quality of the experiment, and its year of publication. A few generic characteristics were common among the studies: the respective devices were used as a transducer to interface between brain activity and a certain psychological, productivity, performance, or interaction characteristic, the number of subjects was relatively small for the human tests, the methodology was rather strict, but the focus was on observing a relationship not on obtaining an accurate measurement. The following paragraphs summarize each one of the four research domains.

Study of Cognition. Given the body of existing research, it is apparent that the primary consumer-grade EEG application is using brainwave signals, once processed through a range of classification algorithms, as a proxy for interpretation of human psychology, intent, mental state, and desire to interact with the system through supervised ML techniques. To support the process, products may integrate brainwave activity with other signals or sensors in order to provide researchers a better holistic picture of the body responds to external stimuli. While popular with a range of applications, consumer-grade devices do not feature in research on the biological mechanisms of human cognitive processing, given their technical limitations, such as the number of channels and sensitivity. This weakness is, however, compensated by their portable nature and user-friendliness, allowing them to be used for mainstream consumer usage rather than in lab conditions and well-trained operators.

Brain–Computer Interface. The launch of consumer-grade EEG devices clearly stimulated the integration of BCI in everyday use. The use of ERP and SSVEP signals expanded from theoretical studies to rehabilitation, including wheelchair control and programming of mobile and web applications. Given the analyzed devices, the products from InteraXon and NeuroSky play just a small part in BCI, most likely due to their small number of built-in channels. In contrast, the electrodes for the Muse and NeuroSky products are more appealing due to the position of their electrodes (located on the frontal region, hence able to collect more frontal lobe information). However, OpenBCI appears to dominate this segment due to its minimalistic circuit board could that can be integrated into any developing system.

Education. One effective EEG application in the field of educational studies was validating and monitoring the effectiveness of traditional and newer learning/teaching methods by observing or promoting attention or developing feedback-based systems to enhance learner's performance. Some consideration was also given to other cognitive functions, such as measurement and tracking of emotion and mental tiredness. However, the most significant contribution of EEG in the education area was related to the online learning platforms, through testing the ability of context reading, either

using adults as test subjects in place of real students in a conventional classroom or, in two separate studies [117,118], observing long-term trends across an entire semester. One other approach was to close the loop between the focus of the audience and adjustments to the delivery style. A startup company BrainCo Inc., supported by Harvard University, USA, proposes one of its products as an EEG headband for classroom tracking (https://www.brainco.tech/) and, through a partnership with a school in China, supplied its products to students and college professors to allow a more informed and flexible relationship during the learning process.

Gaming. Observing the relationship between gaming and the cognitive functions of the players represents a very interesting area of EEG research. In this context, consumer-grade EEG devices have been used to detect emotion, mainly enjoyment, and even to adjust the game conditions in order to improve the engagement with the game and ultimately study and satisfy the user satisfaction. The second avenue of research and development was to use EEG devices as alternative options for navigating and controlling the game. Additionally, this can serve as one of the alternatives for entertainment. It still needs the confirmation of the usage efficiency of these consumer-grade EEG devices in gaming. It should be widely tested by players with special needs in comparison to healthy people to decide that playing games and using these devices would provide equivalent or higher value of entertainment to players in a similar way to traditional methods.

*RQ4*: *Which products were preferred for a certain research topic?* Our study showed that the Emotiv brand is leading psychological research. Emotiv also dominates the gaming-related research topics, which could be explained by the combination of human psychological signals for the study of both cognition and game dynamics. Another brand, NeuroSky, was leading in the field of study of applicability the EEG gadgets to education. In the BCI-related studies that were reviewed, the leader position was divided between OpenBCI and Emotiv products that are popular in use by researchers.

## 6. Conclusions

This paper critically reviews the market of EEG consumer-level products and their suitability and prior usage within studies relating to brain activity. Admittedly, the recently applied low fee, wireless, lightweight, and easy-to-use wearability has established an impact on the increasing attraction towards the noninvasive consumer-grade EEG gadgets among researchers from numerous fields of study. In this study, we summarize both consumer-grade EEG gadgets available in the market and their counterparts, which were used in various research and medical studies, in addition to its usage and reliability. Both traditional applications, i.e., cognition and BCI, and emerging applications, which include educational research and gaming, have been starting to use the devices and demonstrating their validations comparing to the medical-grade EEG devices. Notwithstanding the acceptable capacity and performance of consumer-grade devices, there are grave concerns about the possibility to renew data quality from the smaller quantity of equipped sensors, and at the same time the possible incessant movement because of the portability of the gadgets, which can have resulted in obtained artifacts and influence on its content. Further, it is important to report the products' real capacity and not over-claim the products' actual capability. However, innovative technology increases the development of these consumer-grade EEG gadgets, extending its potential and performing competence. Future efforts could be put on to evaluate its execution through the multidisciplinary collective investigations, with the day-by-day aim to bring it beyond hospitals and laboratories.

**Author Contributions:** M.T. provided the core work for the paper, identifying the relevant studies and aggregating the information, V.S. focused on the communication aspects, cognition, and learning performance, I.K. contributed to the gaming and education sections as well as the research methodology and accuracy of the research sources, S.S. provided support on the education aspects and formatting, and B.G. provided the wider research context and coordinated the paper creation and production. All authors have read and agreed to the published version of the manuscript.

**Funding:** This project has received funding from the European Union's Horizon 2020 research and innovation program under grant agreement no. 833673. This work reflects the author's views, and the agency is not responsible for any use that may be made of the information it contains.

**Conflicts of Interest:** The authors declare no conflict of interest.

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
