# Peer review of "Wireless Sensors for Brain Activity—A Survey"

_electronics, doi:10.3390/electronics9122092_

Round 1

Reviewer 1 Report

The manuscript "Wireless sensors for brain activity - A survey" by TajDini et al. describes the recent literature on the suitability of EEG products in studies related to brain activity. The manuscript documents the recent increase in the use of EEG gadgets besides discussing their utility in research and medical studies. However, certain limitations arising from artifacts due to incessant movements and the requirements to renew data quality from the small number of equipped sensors, etc. still exist and the manuscript discusses these limitations and provides suggestions for future research.

The manuscript is well organized and reports a timely update on this research area. 

Minor:

The discussion section should be expanded with more details about each RQ. 

Author Response

The manuscript "Wireless sensors for brain activity - A survey" by TajDini et al. describes the recent literature on the suitability of EEG products in studies related to brain activity. The manuscript documents the recent increase in the use of EEG gadgets besides discussing their utility in research and medical studies. However, certain limitations arising from artifacts due to incessant movements and the requirements to renew data quality from the small number of equipped sensors, etc. still exist and the manuscript discusses these limitations and provides suggestions for future research.

The manuscript is well organized and reports a timely update on this research area. 

Minor:

The discussion section should be expanded with more details about each RQ

We expanded chapter 5 to include a more detailed discussion of each RQ.

Reviewer 2 Report

Manuscript No.: electronics-992981

Title: Wireless Sensors for Brain Activity – A Survey

Recommendation: Major revision

Overview and general recommendation

This manuscript is a review of wireless sensors used for brain activity and their applicability in four main areas, namely cognition, brain-computer interface, education, and gaming. Furthermore, the authors provide also a sort of guidance for non-medical EEG research. The topic is of interest, since brainwave data are currently used in many application fields, obviously influencing the design of systems and applications related to consumer grade EEG devices.

The paper is written quite well, the English is quite fluent, but it is necessary that authors carefully and thoroughly re-read the paper to correct some typos or grammatical errors (e.g. singular/plural forms, subject-verb agreement, verb tenses, use of commas and prepositions), thus also improving readability (sometimes the presentation is not fully clear).

The contextualization of the study in the literature background quite is good, but some references to MDPI journals, preferably in the context of Electronics, should be added.

There are some style modifications and suggestions provided in the following comments, which may help the authors to improve the paper.

Major comments

Comment #1

Abstract: it seems that the authors have copied different sentences from the manuscript (e.g. lines 41-44, 49-51) and paste them here. However, it is advisable to write the abstract without using the exact sentences used in the text, both to improve readability and to stimulate the reader’s interest.

Section 3: before reporting devices from different vendors, it should be discussed how the performance evaluation is commonly done in the literature. For example, is there s standard test protocols? Which are the parameters usually considered for the comparison? Are there any standards defining minimum performance indices?

A comparison among the different commercial devices should be made, preferably based on quantitative indicators and objective evaluations.

Section 4: more quantitative parameters/indicators describing the different devices performances should be reported, whenever available in the literature, in order to provide the reader with more objective evaluations.

Section 4.1: some references should be added in the first paragraph of this section, to better contextualize the topic. The same for Section 4.3 ad Section 4.4.

Discussion and conclusions sections: some considerations reporting objective parameters describing the performance of the reviewed devices should be added, to allow the reader to choose the most suitable device for an eventual research.

Minor comments

Comment #1

Line 44: the statement “Given the reduced need for standardisation or accuracy” should be justified, or at least a reference should be added.

Line 54: Section 4 is not introduced here, please check.

Lines 90-91: please check this sentence, since it seems that the main clause is missing.

Table 1: it could be interesting to report small images of the different products, if available.

Section 3.1: in order to improve the readability, it could be useful to add a table with the performance accuracies in blinking detection reported in the different literature studies. This new table should include the performance of devices from different vendors, if data are available. Also references related to the studies providing these data should be reported in the table. In this way, the different devices would be more easily comparable for the readers.

Line 152: some quantitative parameters should be reported in relation to the comparison with a medical device. Moreover, is there a protocol widely accepted for the validation of this device or literature works report their own test protocols?

Line 190: the mentioned criteria should be briefly reported, preferably in quantitative terms.

Lines 216-220: on the basis of which parameters the performances are considered “identical” and providing “similar results”?

Lines 232-234: please check this sentence.

Lines 254: what about the accuracy of these devices? Literature provides some data about their performance with respect to medical grade devices?

Table 2: data about accuracy (or other indices classifying the performance) could be reported, where available. Also eventual other sensors used in the studies could be reported. In this way, the table can better summarize the whole section.

The same for the following tables (i.e. Tables 2-5).

Lines 411-412: the sentence should be expanded, specifying that the table refers to the studies that will be described just after.

Lines 414-416: please check this sentence.

Line 433: the improvement of the performance classification should be quantified.

Line 465: if “BLSTMLSTM” is an acronym, it should be explained.

Lines 472-473: this sentence should be moved to the introductory part of Section 4, which could briefly introduce the different subsections.

Line 485: the acronym PCA should be explained.

Line 575-584: these lines should be moved to the introductory part of Section 4.

Line 722: technical limitations should be detailed in quantitative terms.

Lines 765-766: please check this sentence.

References: please note that when reporting a link, the “last date accessed” should be reported.

Funding/acknowledgements: please complete (or remove, if not necessary) these sections.

Author Response

Overview and general recommendation

This manuscript is a review of wireless sensors used for brain activity and their applicability in four main areas, namely cognition, brain-computer interface, education, and gaming. Furthermore, the authors provide also a sort of guidance for non-medical EEG research. The topic is of interest, since brainwave data are currently used in many application fields, obviously influencing the design of systems and applications related to consumer grade EEG devices.

The paper is written quite well, the English is quite fluent, but it is necessary that authors carefully and thoroughly re-read the paper to correct some typos or grammatical errors (e.g. singular/plural forms, subject-verb agreement, verb tenses, use of commas and prepositions), thus also improving readability (sometimes the presentation is not fully clear).

Answer: We identified some typos and grammar errors which were corrected in the updated manuscript

The contextualization of the study in the literature background quite is good, but some references to MDPI journals, preferably in the context of Electronics, should be added.

Answer: We introduced a number of references to MDPI Electronics publications

There are some style modifications and suggestions provided in the following comments, which may help the authors to improve the paper.

Major comments

Comment #1

Abstract: it seems that the authors have copied different sentences from the manuscript (e.g. lines 41-44, 49-51) and paste them here. However, it is advisable to write the abstract without using the exact sentences used in the text, both to improve readability and to stimulate the reader’s interest.

Answer: We rewrote the abstract to improve its readability and minimise similarity with the text in the paper

Section 3: before reporting devices from different vendors, it should be discussed how the performance evaluation is commonly done in the literature. For example, is there s standard test protocols? Which are the parameters usually considered for the comparison? Are there any standards defining minimum performance indices?

A comparison among the different commercial devices should be made, preferably based on quantitative indicators and objective evaluations.

Answer: We acknowledge this is a possible avenue to investigate, but this would expand the paper beyond its current scope and we found no consistent methodology used for comparison. One thread that the studies had in common was the comparison in terms of number of channels recorded and accuracy of readings. To exemplify, we introduced Table 2 that summarises a number of studies looking at the accuracy for NeuroSky.

Section 4: more quantitative parameters/indicators describing the different devices performances should be reported, whenever available in the literature, in order to provide the reader with more objective evaluations.

Answer: We expanded throughout the section the existing tables to present the accuracy observed in the identified studies.

Section 4.1: some references should be added in the first paragraph of this section, to better contextualize the topic. The same for Section 4.3 ad Section 4.4.

Answer: We added additional references to the respective section to improve the research context

Discussion and conclusions sections: some considerations reporting objective parameters describing the performance of the reviewed devices should be added, to allow the reader to choose the most suitable device for an eventual research.

Answer: We agree that, for a specific application, objective parameters can indeed be employed, but the majority of the studies, while investigated a specific application, were device-centric and did not follow a comparative analysis of multiple devices

Minor comments

Comment #1

Line 44: the statement “Given the reduced need for standardisation or accuracy” should be justified, or at least a reference should be added.

Answer: We expanded on the statement and added a reference

Line 54: Section 4 is not introduced here, please check.

Answer: Corrected

Lines 90-91: please check this sentence, since it seems that the main clause is missing.

Answer: Corrected

Table 1: it could be interesting to report small images of the different products, if available.

Answer: We agree, but we are also restricted by space and image copyright issues

Section 3.1: in order to improve the readability, it could be useful to add a table with the performance accuracies in blinking detection reported in the different literature studies. This new table should include the performance of devices from different vendors, if data are available. Also references related to the studies providing these data should be reported in the table. In this way, the different devices would be more easily comparable for the readers.

Answer: We added a table to reflect the performance observed by the studies.

Line 152: some quantitative parameters should be reported in relation to the comparison with a medical device. Moreover, is there a protocol widely accepted for the validation of this device or literature works report their own test protocols?

Answer: We acknowledge this is a possible avenue to investigate, but this would expand the paper beyond its current scope and we found no consistent methodology used for comparison. One thread that the studies had in common was the comparison in terms of number of channels recorded and accuracy of readings.

Line 190: the mentioned criteria should be briefly reported, preferably in quantitative terms.

Answer: We included a summary; the reference is a 26 page document with a detailed enumeration of qualitative criteria and requirements.

Lines 216-220: on the basis of which parameters the performances are considered “identical” and providing “similar results”?

Answer: We provided a summary of the two categories of parameters, the referenced paper includes a rather detailed statistical comparison of the results

Lines 232-234: please check this sentence.

Answer: Corrected

Lines 254: what about the accuracy of these devices? Literature provides some data about their performance with respect to medical grade devices?

Answer: We reduced statement to an assumption

Table 2: data about accuracy (or other indices classifying the performance) could be reported, where available. Also eventual other sensors used in the studies could be reported. In this way, the table can better summarize the whole section.

The same for the following tables (i.e. Tables 2-5).

Answer: We included an accuracy column in the tables, where reported by the respective studies

Lines 411-412: the sentence should be expanded, specifying that the table refers to the studies that will be described just after.

Answer: We rephrased and indicated the respective studies

Lines 414-416: please check this sentence.

Answer: Rephrased

Line 433: the improvement of the performance classification should be quantified.

Answer: Expanded to indicate the accuracy

Line 465: if “BLSTMLSTM” is an acronym, it should be explained.

Answer: The acronym was spelled out

Lines 472-473: this sentence should be moved to the introductory part of Section 4, which could briefly introduce the different subsections.

Answer: Corrected

Line 485: the acronym PCA should be explained.

Answer: The acronym was spelled out

Line 575-584: these lines should be moved to the introductory part of Section 4.

Answer: Corrected

Line 722: technical limitations should be detailed in quantitative terms.

Answer: Phrase was briefly expanded, as the point was discussed earlier on

Lines 765-766: please check this sentence.

Answer: Rephrased

References: please note that when reporting a link, the “last date accessed” should be reported.

Answer: Access dates for online references were pdated

Funding/acknowledgements: please complete (or remove, if not necessary) these sections.

 Answer: Funding completed

Reviewer 3 Report

The manuscript titled 'Wireless Sensors for Brain Activity - A Survey' summarizes the devices and research projects in non-medical EEG studies.
The text is important and interesting.

(1) In my opinion, the text could be shorter. For example, the historical relationships between devices shown in section 3 (lines 85-266) may be deleted, or moved to additional materials. The text is about devices and studies using these devices, therefore the description of disorders are not necessary here. please review the text carefully and remove the parts not related to the topic. This manuscript is not collection of abstracts of articles focused on non-medical usage of EEG.

(2) Table 1 provides a good summary of the EEG product.
The most important issue for section 3 is that the most important device parameters like accuracy, error rate etc. are only available inside paragraphs. I definitely propose an additional table with these parameters, perhaps with a note that the values ​​are suggested by the manufacturers. Ideally if they are measured on the same reference data set, but probably it is unpossible in this work.

(3) The analyzes are consistent, but I suggest modifying the parameters provided. The tables provide the sampling rate, but it is not the most important parameter of the depicted projects. In addition, the sampling frequency is given in Table 1. Please remove it.

(4) What is interesting is the obtained quality in experiments, for example accuracy or error, sensitivity or specificity, F1, preferably all mentioned above. Please add such numbers. In my opinion this is the most important issue. Problem (4) applies to Table 2, 3, 4, maybe Table 5.

Author Response

The manuscript titled 'Wireless Sensors for Brain Activity - A Survey' summarizes the devices and research projects in non-medical EEG studies.
The text is important and interesting.

(1) In my opinion, the text could be shorter. For example, the historical relationships between devices shown in section 3 (lines 85-266) may be deleted, or moved to additional materials. The text is about devices and studies using these devices, therefore the description of disorders are not necessary here. please review the text carefully and remove the parts not related to the topic. This manuscript is not collection of abstracts of articles focused on non-medical usage of EEG.

Answer: We aimed to provide a brief historical context of the devices, given their introduction to the EEG context was not research-driven, but market-imposed.

(2) Table 1 provides a good summary of the EEG product.
The most important issue for section 3 is that the most important device parameters like accuracy, error rate etc. are only available inside paragraphs. I definitely propose an additional table with these parameters, perhaps with a note that the values ​​are suggested by the manufacturers. Ideally if they are measured on the same reference data set, but probably it is unpossible in this work.

Answer: We agree with the comment and tried to answer it, subject to the limited accuracy data provided by the respective studied. The tables in section 3 were expanded therefore to include accuracy data, where available

(3) The analyzes are consistent, but I suggest modifying the parameters provided. The tables provide the sampling rate, but it is not the most important parameter of the depicted projects. In addition, the sampling frequency is given in Table 1. Please remove it.

Answer: We do agree with the comment, sampling frequency was removed

(4) What is interesting is the obtained quality in experiments, for example accuracy or error, sensitivity or specificity, F1, preferably all mentioned above. Please add such numbers. In my opinion this is the most important issue. Problem (4) applies to Table 2, 3, 4, maybe Table 5.

Answer: As highlighted above, we aimed to include as much information as possible, given the limited quantitative information in the identified studies

Round 2

Reviewer 2 Report

The authors have improved the paper quality according to most of my previous recommendations, so in my opinion the paper is suitable for publication in Electronics.

Author Response

We thank the reviewer for his/her valuable comments and the time spend helping us improve the paper. 

Reviewer 3 Report

Some of my comments were properly addressed, thank you.
Please consider still unresolved issues:

(1) All tables should use the product names from Table 1. For example, in Table 3 and Table 6 there is an entry "NeuroSky Mindwave Mobile" which is not listed in Table 1.
(2) Moreover, the device names could have unique and short text identifiers that will be used in other tables.

(3) The quality of the experiments presented in each paper is very important, therefore it is not appropriate to present the value of "n / a" in summary. Please review the manuscript carefully to find the experiments quality as a number. If there is no accuracy directly, you can calculate this value using the reported values like error, sensitivity, specificity. In the revised manuscript, Table 5 and Table 6 do not include qualitative results for the methods presented.

Author Response

  • All tables should use the product names from Table 1. For example, in Table 3 and Table 6 there is an entry "NeuroSky Mindwave Mobile" which is not listed in Table 1.

Answer: We corrected the issue

  • Moreover, the device names could have unique and short text identifiers that will be used in other tables.

Answer: We agree that this would slightly reduce the text, but it would make the paper slightly more cryptic, as the names are recognised in the long form, as they appear in the paper

  • The quality of the experiments presented in each paper is very important, therefore it is not appropriate to present the value of "n / a" in summary. Please review the manuscript carefully to find the experiments quality as a number. If there is no accuracy directly, you can calculate this value using the reported values like error, sensitivity, specificity. In the revised manuscript, Table 5 and Table 6 do not include qualitative results for the methods presented.

Answer: We agree this is not ideal, but the insertion of specific values was a request from other reviewers; a significant proportion of the papers included qualitative results and therefore, while recognisable as prior research, they did not include actual figures